# Chronic Hepatitis E in Rheumatology and Internal Medicine Patients: A Retrospective Multicenter European Cohort Study

**DOI:** 10.3390/v11020186

**Published:** 2019-02-22

**Authors:** Sven Pischke, Jean-Marie Peron, Moritz von Wulffen, Johann von Felden, Christoph Höner zu Siederdissen, Sophie Fournier, Marc Lütgehetmann, Christoph Iking-Konert, Dominik Bettinger, Gabriella Par, Robert Thimme, Alain Cantagrel, Ansgar W. Lohse, Heiner Wedemeyer, Robert de Man, Vincent Mallet

**Affiliations:** 1Department of Medicine, University Medical Centre Hamburg-Eppendorf, 20246 Hamburg, Germany; s.pischke@uke.de (S.P.); m.von-wulffen@uke.de (M.v.W.); j.von-felden@uke.de (J.v.F.); a.lohse@uke.de (A.W.L.); 2German Centre for Infection Research (DZIF), Hamburg partner site, 20246 Hamburg, Germany; mluetgehetmann@uke.de (M.L.); Wedemeyer.Heiner@mh-hannover.de (H.W.); 3Service d’hépato-gastroentérologie, Hôpital Purpan CHU Toulouse, Université Paul Sabatier III, 31000 Toulouse, France; peron.jm@chu-toulouse.fr (J.-M.P.); fournier.s@chu-toulouse.fr (S.F.); cantagrel.a@chu-toulouse.fr (A.C.); 4Gastroenterology, Hannover Medical School, 30625 Hannover, Germany; HoenerzuSiederdissen.Christoph@mh-hannover.de; 5Microbiology, University Medical Centre Hamburg-Eppendorf, 20246 Hamburg, Germany; 6Rheumatology, University Medical Centre Hamburg-Eppendorf, 20246 Hamburg, Germany; c.iking-konert@uke.de; 7Department of Medicine II, Medical Center University of Freiburg, Faculty of Medicine, University of Freiburg, Hugstetter Str. 55, 79106 Freiburg, Germany; dominik.bettinger@uniklinik-freiburg.de (D.B.); robert.thimme@uniklinik-freiburg.de (R.T.); 8Berta-Ottenstein-Programme, Faculty of Medicine, University of Freiburg, 79106 Freiburg, Germany; 9Clinical Centre, First Department of Medicine, University of Pécs, H-7622 Pécs, Hungary; pargabriella@gmail.com; 10Gastroenterology and Hepatology, Erasmus MC Medical Center Rotterdam, 3062 PA Rotterdam, The Netherlands; r.deman@erasmusmc.nl; 11Université Paris Descartes, Assistance Publique—Hôpitaux de Paris (AP—HP), Hôpital Cochin, Hepatology Service, Institut National de la Santé et de la Recherche Médicale unité 1223, Institut Pasteur, 75006 Paris, France

**Keywords:** hepatitis E, chronic hepatitis E, disease-modifying antirheumatic drugs (DMARDs), ribavirin

## Abstract

**Objectives:** Hepatitis E virus (HEV) infection is a pandemic with regional outbreaks, including in industrialized countries. HEV infection is usually self-limiting but can progress to chronic hepatitis E in transplant recipients and HIV-infected patients. Whether other immunocompromised hosts, including rheumatology and internal medicine patients, are at risk of developing chronic HEV infection is unclear. **Methods:** We conducted a retrospective European multicenter cohort study involving 21 rheumatology and internal medicine patients with HEV infection between April 2014 and April 2016. The underlying diseases included rheumatoid arthritis (*n* = 5), psoriatic arthritis (*n* = 4), other variants of chronic arthritis (*n* = 4), primary immunodeficiency (*n* = 3), systemic granulomatosis (*n* = 2), lupus erythematosus (*n* = 1), Erdheim–Chester disease (*n* = 1), and retroperitoneal fibrosis (*n* = 1). **Results:** HEV infection lasting longer than 3 months was observed in seven (33%) patients, including two (40%) patients with rheumatoid arthritis, three (100%) patients with primary immunodeficiency, one (100%) patient with retroperitoneal fibrosis and one (100%) patient with systemic granulomatosis. Patients with HEV infection lasting longer than 3 months were treated with methotrexate without corticosteroids (*n* = 2), mycophenolate mofetil/prednisone (*n* = 1), and sirolimus/prednisone (*n* = 1). Overall, 8/21 (38%) and 11/21 (52%) patients cleared HEV with and without ribavirin treatment, respectively. One patient experienced an HEV relapse after initially successful ribavirin therapy. One patient (5%) was lost to follow-up, and no patients died from hepatic complications. **Conclusion:** Rheumatology and internal medicine patients, including patients treated with methotrexate without corticosteroids, are at risk of developing chronic HEV infection. Rheumatology and internal medicine patients with abnormal liver tests should be screened for HEV infection.

## 1. Introduction

Hepatitis E virus (HEV), which is the causative agent of hepatitis E, is a non-enveloped RNA virus and a member of the genus *Orthohepevirus* within the Hepeviridae family. There are four major HEV genotypes (1, 2, 3, and 4) that can infect humans. The preferential mode of transmission of HEV is enteric via drinking water contaminated by HEV-1 or HEV-2 or via infected food contaminated by HEV-3 or HEV-4. HEV infection is a pandemic, including in high-income countries, where HEV-3 is the most prevalent genotype [1].

In general, HEV infection is self-limiting but can evolve to chronic hepatitis E in immunocompromised patients, including transplant recipients and HIV-infected patients [2,3]. In solid organ recipients, chronic HEV infection is defined as the persistence of viral replication beyond 3 months after infection [4]. In general, patients with chronic hepatitis E clear HEV infection when immunity is restored or a short course of ribavirin monotherapy is administered [5,6]. 

The HEV infection course in internal medicine/rheumatology patients has been infrequently reported. Whether internal medicine/rheumatology patients, including patients treated with disease-modifying agents (DMARDs), are at risk of developing chronic hepatitis E remains controversial [7,8].

We analyzed the course of HEV infection in a retrospective cohort from seven internal medicine and rheumatology centers across Europe and show that internal medicine/rheumatology patients, including patients treated with DMARDs, are at risk of developing chronic hepatitis E.

## 2. Patients and Methods

### 2.1. Patients

We conducted a retrospective European case series study that included data from 7 rheumatology and internal medicine centers in Germany, Italy, the United Kingdom, the Netherlands and France. The data included all cases of HEV infection diagnosed between April 2014 and April 2016. An investigator from each center collected the patient data from the medical records. All authors vouch for the completeness and accuracy of the presented data. 

Because of the retrospective and observational nature of the study and in accordance with German law, ethical approval for the study was waived. Some patients (*n* = 9) were previously included in two case series [7,8].

The decision to lower the patients’ immunosuppressive regimen or treat patients with ribavirin was made individually based on the physicians’ experience. The dosing and duration of the ribavirin treatment were not standardized.

### 2.2. Virological Assessment

In all patients, HEV infection was confirmed by nucleic acid testing (NAT). In 18 patients polymerase chain reaction (PCR) positivity based on in-house PCR assays, while in 3 patients a commercial PCR assay was used (Altona Diagnostics, Hamburg, Germany). The majority of used PCR assays had a lower limit of detection (95% hit rate) of 143 IU/mL as determined by the first World Health Organization (WHO) standard for HEV RNA nucleic acid amplification testing-based assays. The proven duration of infection was determined prospectively or retrospectively based on stored frozen serum samples. A sustained virological response was defined as an undetectable level of HEV RNA in the serum 24 weeks after the completion of ribavirin therapy. 

To confirm PCR results serological testing was performed in all patients. In the majority (*n* = 14) the Wantai anti HEV IgG assay (Wantai, Bejing, China) was used according to manufacturers instructions.

## 3. Results

### 3.1. Cohort Characteristics

The cohort comprised 21 patients (Table 1). The sex ratio was balanced (*n* = 11 [52%] males). The age of the patients ranged from 25 to 75 years (median 57 years). All patients presented with detectable anti HEV IgG at diagnosis ot within the further course. The underlying diseases included rheumatoid arthritis (*n* = 5), psoriatic arthritis (*n* = 4), other variants of chronic arthritis (*n* = 4), primary immunodeficiency (*n* = 3), systemic granulomatosis (*n* = 2), lupus erythematosus (*n* = 1), Erdheim–Chester disease (*n* = 1), and retroperitoneal fibrosis (*n* = 1). 

We could not determine the exact ratio of HEV positive patients in the total cohort of internal medicine/rheumatology patients. In total, 3,800 patients are treated annually at the Rheumatological Department of the University Hospital Hamburg Eppendorf and the clinic at Bad Bramstedt. The total number (percent) of patients diagnosed with HEV in this subgroup was five (0.07%) during the observational study period. 

The immunosuppressive treatments included methotrexate monotherapy (*n* = 4), anti-TNF (tumor necrosis factor) monotherapy (*n* = 4), methotrexate/anti-TNF combination therapy (*n* = 4), methotrexate/rituximab (*n* = 1), methotrexate/prednisolone (*n* = 1), sirolimus/prednisolone (*n* = 1), mycophenolate mofetil/prednisolone (*n* = 1), abatacept (*n* = 1), and cyclophosphamide (*n* = 1). 

All but one of the HEV infections were locally acquired. HEV genotype 3c was identified in two patients, and HEV genotype 3f was identified in 2 patients. One case (genotype 1) was imported from Indonesia (patient #5, Table 1). This patient was a 65-year-old woman with rheumatoid arthritis. She cleared the infection without any relevant problems after the cessation of methotrexate. 

In most patients, HEV genotyping was not performed.

The HEV viral load was determined in 10 patients and ranged from 216 IU/mL to 8,600,000 IU/mL (mean 1,123,183 IU/mL, standard deviation 2,653,320 IU/mL). 

The peak values of ALT (Alnanine amino transferasis)ranged from 88 to 5231 U/L (median 610 U/L) during the HEV infection, while the ALT levels ranged from 15 to 35 U/L (median 22 U/L, missing: *n* = 11) before the HEV infection and ranged from 4–47 U/L (median 17 U/L, missing: *n* = 9) after the HEV infection (*p* < 0.001, *p* < 0.001, Figure 1).

### 3.2. Course of Hepatitis E Virus (HEV) Infection

HEV RNA was detected by NAT in all patients at the time of the diagnosis of HEV infection. The duration of the HEV infection ranged from 0 to 96 weeks (median 4 weeks). No liver-related complications, including cirrhosis and end-stage liver disease, were reported. No extrahepatic manifestation attributable to the HEV infection was observed [9]. Seven patients developed a course of HEV infection lasting longer than 3 months (Table 1, Figure 1). Three patients with chronic HEV infection had a hereditary impairment of their immune response (patients #3, 10, and 20; Table 1). One patient with chronic HEV infection had retroperitoneal fibrosis and was treated with sirolimus/prednisolone. One patient with chronic HEV infection (#18) had systemic granulomatosis and was treated with mycophenolate mofetil/prednisolone. The remaining two patients with chronic HEV infection had rheumatoid arthritis (#6 and #11) and were treated with either methotrexate or abatacept.

### 3.3. Treatment of Chronic HEV Infection

The total cohort can be divided into the following 4 subgroups (Table 2): patients who spontaneously cleared the infection, patients who cleared the infection after a reduction of immunosuppression, patients who have been treated with ribavirin and patients who have been treated with a reduction of immunosuppression plus ribavirin. The duration of the HEV infection and the ALT peak levels did not significantly differ between these groups (*p* = ns) (Figure 1).

Ribavirin treatment (11–13.5 mg/kg body weight, median 12.8 mg/kg; duration 1–26 weeks, median 12 weeks) was initiated in 9 patients (7 ribavirin mono therapy), including 5 patients with chronic HEV infection (#6, #10, #18, #19, and #20; Table 1). Ribavirin dose reduction was only necessary in one patient. No serious adverse events resulting from the ribavirin treatment were reported. Immunosuppressive treatment was alleviated or differed in 8 (38%) patients, including two patients simultaneously treated with ribavirin. Abatacept therapy was stopped in 1 patient (#11) with chronic HEV infection. HEV infection did not lead to any treatment modification in 6 (29%) patients. All patients achieved a sustained virological response (one patient lost to follow up).

Only one of the patients (#10) relapsed after the ribavirin therapy. No patient relapsed in the group without ribavirin therapy. 

The ALT peak and duration of viremia did not significantly differ among the patients without a change in immunosuppression, patients with reduced immunosuppression and patients treated with ribavirin (Figure 1, *p* = ns).

### 3.4. Outcome of Selected Patients

No patients died from hepatic complications. One patient with chronic infection (#3) was lost to follow-up. The outcomes of the seven remaining patients with chronic infection are described below. The lymphocyte count was determined in 5 patients at the time of HEV infection and ranged from 560–1900 cells/mm^3^ (median: 1050 cells/mm^3^). The total IgG level was determined in 5 patients and ranged from 7.6–14.1 g/dL (median 11.3 g/dL).

Patient #3 was a 51-year-old male with common variable immunodeficiency and chronic HEV infection lasting for more than 6 months. He was lost to follow-up.Patient #6 was a 75-year-old female under methotrexate treatment for rheumatoid arthritis with chronic hepatitis E lasting for more than 4 months. Her INR and total bilirubin peaked at 1.4 and 25.6 mg/dL (437 µmol/L), respectively. She experienced clearance of the HEV infection after 26 weeks of ribavirin treatment.Patient #10 was a woman aged 57 years with idiopathic CD4 T lymphocytopenia and a primary deficiency in IgG-1, -2, and -4 subclasses. Total lymphocyte count was 0.41 109 cells/L. The CD4 T-lymphocyte count was 0.22 109 cells/L. The CD8 T-lymphocyte count was 0.05 109 cells/L. The CD19 B-lymphocyte count was 0.03 109 cells/L. Chronic hepatitis E was diagnosed in December 2009. Ribavirin treatment 600 mg/d (12 mg/kg) was initiated for 12 weeks. The patient relapsed after treatment. Ribavirin was reintroduced without a virological response. Ribavirin was pursued at lower doses to maintain normal liver function tests. The patient died under ribavirin with metastatic epidermoidal cancer in 2018.Patient #11 was a 51-year-old female under abatacept treatment for rheumatoid arthritis with chronic HEV infection lasting for more than 16 weeks. She exhibited clearance of the HEV infection after withdrawal from abatacept.Patient #18 was a 29-year-old male under mycophenolate (1500 mg daily) and prednisolone (7.5 mg daily) treatment for systemic granulomatosis with chronic HEV infection lasting for more than 1 year. He exhibited clearance of the HEV infection after 5 months of ribavirin treatment.Patient #19 was a 34-year-old male under prednisolone (60 mg daily) and sirolimus (4 mg daily) treatment for retroperitoneal fibrosis with chronic HEV infection lasting for more than 2 years. He exhibited clearance of the HEV infection after 5 months of ribavirin treatment.Patient #20 was a 48-year-old female with an undefined CD4 cell deficiency and chronic HEV infection. She exhibited clearance of the HEV infection after 5 months of ribavirin treatment.

## 4. Discussion

In a multicenter, international cohort of 21 internal medicine/rheumatology patients, HEV infection persisted for more than 12 weeks in 7 (33.3%) patients and more than 24 weeks in 5 (24%) patients (Table 1). Chronic hepatitis E was associated with methotrexate or abatacept treatment in two rheumatoid arthritis patients. All but one of the patients cleared HEV infection (one further patient was lost to follow-up), including 15 (71%) patients who cleared the infection after the discontinuation of treatment with immunosuppressants and/or ribavirin. The ALT peak levels and duration of HEV infection did not significantly differ among patients treated with various antiviral strategies, such as ribavirin or reduction of immunosuppression (Figure 1). Thus, the efficacy of these regimens could not be evaluated, and larger cohort studies are needed.

Our findings demonstrate that the progression of HEV infection from acute to chronic is not limited to organ transplant recipients and HIV-infected patients with a low CD4 T cell count [10,11,12] and can occur in internal medicine/rheumatology patients, including patients undergoing treatment with DMARDs. Unexpectedly, we observed chronic HEV infection in a patient undergoing methotrexate treatment for rheumatoid arthritis without overt lymphopenia. In contrast to our findings, a previous French observational study did not report any case of chronic HEV infection among rheumatology patients, including rheumatoid arthritis patients [7]. However, both studies are retrospective analyses of small unstructured cohorts. Perhaps cessation of immunosuppression or initiation of ribavirin interfered with the clinical course of HEV infection. It is still completely unclear, how many patients might have developed chronic infection without any actions of the responsible physicians.

Other case reports of chronic hepatitis E among patients with immunological/rheumatological diseases were limited to strongly immunosuppressed patients [13,14]. Chronic HEV infection has been associated with lymphopenia, a low CD4 T cell count and impaired HEV-specific T cell response [15]. The retrospective nature of our study did not allow us to identify the risk factors for chronic HEV infection among the internal medicine/rheumatology patients. Our clinical study does not warrant any insights into the underlying pathophysiology of chronic HEV infection in rheumatological patients. However, recently a study using a pig model revealed that active suppression of cell-mediated immune responses under immunocompromised conditions may facilitate the establishment of chronic HEV infection [16]. Further studies investigating the exact role of methotrexate or anti-TNF medications in this context are needed. 

Our study also highlights the relevance of HEV infection as a potential differential diagnosis in internal medicine/rheumatology patients with elevated transaminase levels. HEV is an endemic worldwide, including in industrialized countries [1]. In general, internal medicine/rheumatology patients are treated as outpatients, and the diagnosis of chronic hepatitis E can easily be overlooked; at least initially, patients may have no symptoms. Often, the only clues regarding the diagnosis are mild fluctuating abnormalities in liver function tests. These abnormalities are commonly and incorrectly attributed to drug-induced liver injury given the polypharmacy that many patients receive. In addition, serological screening in these types of patients often yields false-negative results [17]. The diagnosis requires the demonstration of the persistence of HEV RNA by NAT. 

However, while more than 1% (*n* = 4) of 287 liver transplant recipients at the University Hospital Hamburg Eppendorf presented with hepatitis E [18], only 0.07% of rheumatological patients at the same center were diagnosed with HEV infection in the present study. These data indicate that HEV infections in rheumatological patients rarely develop chronic courses.

All but one chronic hepatitis E patients cleared the HEV infection after ribavirin treatment with or without the discontinuation of immunosuppressive treatment. The high rate of sustained virological response is consistent with the results of previous studies in other populations of immunosuppressed patients [6]. Because of the uncontrolled nature of our study, we cannot exclude the possibility that some patients could have resolved the HEV infection after the discontinuation of immunosuppressive treatment without ribavirin treatment. We also cannot exclude the possibility that some patients could have cleared the HEV infection at a time beyond 12 to 24 weeks. In a previous study, no HEV clearance was observed in solid organ transplant patients between months 3 and 6 after infection [4]. The burden of chronic HEV infection, including cirrhosis and extrahepatic manifestations, has been reported in immunosuppressed patients [2,19]. Evidence supporting the guidelines for the treatment of immunosuppressed patients with chronic HEV infection, including the alleviation of immunosuppressive treatment and the provision of antiviral treatment, is increasing [20]. Fortunately, the European Association for the Study of the Liver (EASL) guidelines for hepatitis E have been released this year and provide some critical recommendations for HEV in transplant recipients. The current knowledge regarding HEV infections in immunosuppressed patients other than transplant recipients and HIV patients is still rudimentary. The present study largely contributes to this field. The EASL guidelines clearly suggest treating HEV infection in immunosuppressed patients lasting for more than 3 months either by decreasing immunosuppression or with ribavirin [21]. Our observations demonstrate that both the cessation of immunosuppression and ribavirin can be used safely in rheumatological patients. 

As in rheumatological patients, knowledge regarding HEV is still limited in patients with hematological diseases and patients with inflammatory bowel diseases. Recently, two case reports on HEV infection in inflammatory bowel disease patients have been published. One case of spontaneously self-limited HEV infection in a patient with ulcerative colitis [22] and one uncommon case of chronic HEV genotype 1 infection in a patient with Crohn’s disease treated with mercaptopurine, adalimumab and prednisolone [23] have been reported. 

The present study depicts a large cohort of a rare phenomenon. However, this retrospective analysis has several limitations, as follows: the viral load and the genotype of each patient were not determined. Some clinical data are missing due to the retrospective, unstandardized characteristics of the study, including a standardized assessment of hepatic and extrahepatic symptoms. Prospective, standardized multicentric cohort studies are needed to optimize our understanding of the relevance of HEV infections in rheumatological and internal medicine patients.

In conclusion, HEV should always be a part of the workup for internal medicine/rheumatology patients with abnormal liver function tests. Internal medicine/rheumatology patients, including patients under methotrexate treatment, are at risk of developing chronic HEV infection. The alleviation of immunosuppressive treatment with or without ribavirin treatment should be considered for internal medicine/rheumatology patients with chronic hepatitis E. Future studies are needed to describe the burden of chronic hepatitis E among IM/R patients.

## Figures and Tables

**Figure 1 viruses-11-00186-f001:**
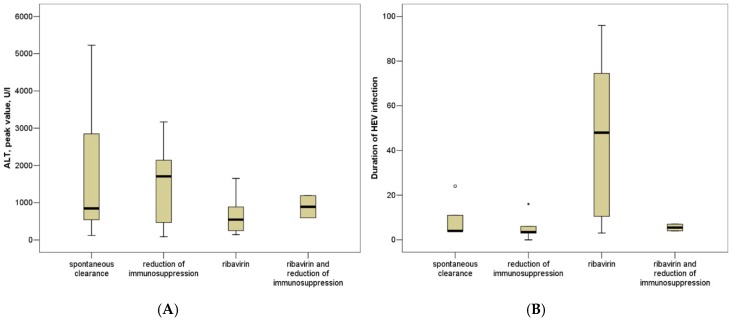
ALT peak levels (**A**) and duration of proven HEV viremia (**B**) in patients with spontaneous clearance, patients with reduction of immunosuppression, patients treated with ribavirin and patients treated with reduction of immunosuppression plus ribavirin.

**Table 1 viruses-11-00186-t001:** Characteristics of internal medicine/rheumatology patients with hepatitis E virus (HEV) infection (chronic patients with more than 3 months of viremia: gray shading).

Patient	Underlying Disease	Sex	Age (yrs)	Peak ALT (IU/mL)	Treatment for Rheumatic Disease	Duration of Viremia (weeks)	Intervention
#1	Rheumatoid arthritis	M	57	1081	Methotrexate, rituximab	11	No intervention
#2	Rheumatoid arthritis	F	69	610	Methotrexate, anti-TNF	4	No intervention
#3	*CVID*	M	57	2849	No immunosuppressive therapy	> 24	No intervention, lost to follow-up
#4	Systemic lupus erythematosus	F	56	543	Methotrexate, anti-TNF	4	No intervention
#5	Rheumatoid arthritis	F	65	469	Methotrexate	4	Discontinuation of immunosuppression
#6	Rheumatoid arthritis	F	75	1654	Methotrexate	18	Ribavirin treatment
#7	Psoriatic arthritis	M	67	1201	Anti-TNF biotherapy	11	Ribavirin treatment
#8	Erdheim–Chester disease	M	58	2140	Methotrexate, anti-TNF	3	Discontinuation of immunosuppression
#9	Granulomatosis	M	59	5231	Cyclophosphamide	4	No intervention
#10	Primary immune deficiency	F	57	546	No immunosuppressive therapy	53	Ribavirin treatment (relapse)
#11	Rheumatoid arthritis	F	51	1750	Abatacept	16	Discontinuation of immunosuppression
#12	Juvenile arthritis	F	30	591	Methotrexate, anti-TNF	4	Ribavirin, discontinuation of IS
#13	Psoriatic arthritis	F	54	121	Anti-TNF	4	No intervention
#14	Psoriatic arthritis	M	62	1190	Methotrexate	7	Ribavirin, discontinuation of IS
#15	Axial spondyloarthritis	F	52	142	Infliximab	3	discontinuation of IS
#16	Psoriatic arthritis	F	25	88	Methotrexate, anti-TNF	< 1	Discontinuation of immunosuppression
#17	Undetermined arthritis	M	70	3170	Methotrexate/Prednisolone	6	Discontinuation of immunosuppression
#18	Granulomatosis	M	29	216	Mycophenolate/prednisolone	48	Ribavirin treatment
#19	Retroperitoneal fibrosis	M	34	568	Sirolimus/prednisolone	96	Ribavirin treatment
#20	Undefined CD4 disturbance	M	48	282	No immunosuppressive therapy	96	Ribavirin treatment
#21	Psoriatic arthritis	M	55	1669	Anti-TNF	3	Discontinuation of immunosuppression

**Table 2 viruses-11-00186-t002:** Comparison of patient cohorts.

	Patients without Antiviral Treatment (*n* = 6)	Patients with Reduced Immunosuppression (*n* = 6)	Patients Treated with Ribavirin (*n* = 7)	Patients Treated with Reduced Immunosuppression Plus Ribavirin (*n* = 2)
Male	3 (50%)	3 (50%)	4 (57%)	1 (50%)
Age in years, mean (range, SD)	59 (54–69, 5)	54 (25–70, 16)	52 (29–75, 17)	46 (30–62)
Bilirubin peak, mean (range, SD), mg/dL	4.1 (1.0–8.0, 2.9)	1.3 (1.0–1.5, 0.4)	3.5 (1.7–7.4, 2.5)	1.0 (1.0–1.0, nd)
ALT peak in U/L	1739 (121–5231, 1959)	1548 (88–3170, 1125)	658 (142–1654, 564)	891 (591–1190, 424))
Duration of proven HEV viremia in weeks (range, SD)	9 (4–24, 8)	5 (0–16, 6)	45 (3–96, 40)	6 (4–7, 2)
Underlying diseases	-> 2 Rheumatoid a. -> 1 Psoriatic a. -> 1 Granulomatosis -> 1 CVID -> 1SLE	-> 2 Rheumatoid a. -> 2 Psoriatic a. -> 1 Erdheim Chester -> 1 Undefined a.	-> 1 Rheumatoid a. -> 1 Psoriatic a. -> Axial spondyloarthritis -> 1 Granulomatosis -> Primary immune deficiency -> Retroperitoneal fibrosis -> Undefined CD4 disturbance	-> 1 Psoriatic a. -> Juvenile

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
