# Peer review of "Chronic Hepatitis E in Rheumatology and Internal Medicine Patients: A Retrospective Multicenter European Cohort Study"

_viruses, 2019, doi:10.3390/v11020186_

Round 1

Reviewer 1 Report

In this manuscript (Viruses-437911), authors analyzed in a retrospective cohort from seven internal medicine and rheumatology centers across Europe and show that internal medicine/rheumatology patients,including patients treated with DMARDs, are at risk of developing chronic hepatitis E.

Comments:

(1)  Page 3, Section Virological assessment: Authors should expand this section. They should indicate the test used for the detection of HEV RNA. Were IgM and IgG antibodies detected? What test did they use to detect antibodies? What sensitivity and specificity did these tests have? Have they genotyped? How was HEV the genotyping performed? All these data should be included in this section because some of these data are indicated in the results, for example: page 3, line 118: "In most of patients, HEV genotyping was not performed".

Author Response

Reviewer 1:

Comments and Suggestions for Authors
In this manuscript (Viruses-437911), authors analyzed in a retrospective cohort from seven internal medicine and rheumatology centers across Europe and show that internal medicine/rheumatology patients,including patients treated with DMARDs, are at risk of developing chronic hepatitis E.

è We thank this reviewer for reviewing our manuscript and for the helpful comments. We underlined changes in the revised version in yellow colour.

Comments:

(1)  Page 3, Section Virological assessment: Authors should expand this section. They should indicate the test used for the detection of HEV RNA. Were IgM and IgG antibodies detected? What test did they use to detect antibodies? What sensitivity and specificity did these tests have? Have they genotyped? How was HEV the genotyping performed? All these data should be included in this section because some of these data are indicated in the results, for example: page 3, line 118: "In most of patients, HEV genotyping was not performed".

è We added the PCR technique used to the methods and added a sentence regarding serology to the methods and to the results section.

Reviewer 2 Report

The manuscript entitled "Chronic hepatitis E in rheumatology and internal medicine patients: a retrospective multicenter European cohort study" described cases of HEV infection in a population of immunosuppressed rheumatology or internal medicine patients. This retrospective study includes 21 HEV-positive patients from 7 clinical units of 5 different European countries. This study shows that 7 out of the 21 patients became chronically-infected and that all except one could be successfully cured either with ribavirin treatment or/and discontinuation of immunosuppression. In 9 cases, a therapeutical option, i.e. ribavirin or adjustment of immunosuppression, was given before observing a viral persistence (more than 3 months of viremia). Overall, the compilation of cases shows that patients other than those with solid-organ transplant, HIV or hematological cancer are also at risk to develop chronic hepatitis E. While this notion was already known, this retrospective study provides an interesting summary of what is observed in Europe nowadays. This report provides the rationale for better documented prospective studies in the future.

Minor issue:

- Table 1, Patient #16: The duration of viremia is 0. Is it a typing mistake or is it that the duration could not be determined ? In the latter case, it would be better to indicate "n.d."

- Table 1, Duration of viremia: A single value is given for the duration of viremia. How is the viremia duration determined ? Is there so frequent blood samples to give an exact number of weeks ? Would it not be better to provide a range of weeks ? As it is an important parameter for the study, this should be clarified, e.g. with changes in the Table or explanations in the Methods.

- p7 l177: “Patient #10 was a white woman...“, Ethnic parameters were not mentioned for the other patients and are probably not relevant to the study, therefore I would omit this information here.

- p7 l208: “regiments“ should read “regimens“

Author Response

Reviewer 2:

Comments and Suggestions for Authors
The manuscript entitled "Chronic hepatitis E in rheumatology and internal medicine patients: a retrospective multicenter European cohort study" described cases of HEV infection in a population of immunosuppressed rheumatology or internal medicine patients. This retrospective study includes 21 HEV-positive patients from 7 clinical units of 5 different European countries. This study shows that 7 out of the 21 patients became chronically-infected and that all except one could be successfully cured either with ribavirin treatment or/and discontinuation of immunosuppression. In 9 cases, a therapeutical option, i.e. ribavirin or adjustment of immunosuppression, was given before observing a viral persistence (more than 3 months of viremia). Overall, the compilation of cases shows that patients other than those with solid-organ transplant, HIV or hematological cancer are also at risk to develop chronic hepatitis E. While this notion was already known, this retrospective study provides an interesting summary of what is observed in Europe nowadays. This report provides the rationale for better documented prospective studies in the future.

è We thank the reviewer for his helpful comments, which helped to clarify some aspects.

Minor issue:
- Table 1, Patient #16: The duration of viremia is 0. Is it a typing mistake or is it that the duration could not be determined ? In the latter case, it would be better to indicate "n.d."

è We changed into less than 1 (<1)
- Table 1, Duration of viremia: A single value is given for the duration of viremia. How is the viremia duration determined ? Is there so frequent blood samples to give an exact number of weeks ? Would it not be better to provide a range of weeks ? As it is an important parameter for the study, this should be clarified, e.g. with changes in the Table or explanations in the Methods.

è The reviewer is right. We analysed the “proven” duration of viremia. This means the time period of viremia basing on positive PCR results. We added the term “proven” to clarify this.

- p7 l177: “Patient #10 was a white woman...“, Ethnic parameters were not mentioned for the other patients and are probably not relevant to the study, therefore I would omit this information here.

è This is right, we deleted the word “white”

- p7 l208: “regiments“ should read “regimens“

è We changed this typo, thanks.

Reviewer 3 Report

In order to investigate the risk of chronic viral replication in immunocompromised patients other than transplant recipients and HIV infected patients. The authors of this manuscript examined the course of HEV infection in a retrospective cohort from seven internal medicine and rheumatology centers across Europe between 2014 and 2016. The result showed that 7 out of 21 rheumatology and internal medicine patients with HEV infection developed chronic HEV infection. They concluded that internal medicine/rheumatology patients, including patients treated with DMARDs, are at risk of developing chronic hepatitis E. Although there were publications on the same topic, this manuscript utilized more data and presented a more convincible conclusion. The study were careful carried out, and the manuscript is well written. However, I have two comments would like the authors to address.

1) The Ref. 7 in this manuscript, suggested that there is no risk of chronicity of this infection in patients with inflammatory arthritis treated with immunosuppressants (Bauer H, et al. Medicine (Baltimore). 2015;94(14):e675.), which is conflict with the conclusion of this study. It is noteworthy that the outcome of the HEV infection in the previous study is under the interference of treatment. How long had the HEV infection been before the diagnosis? What happen if there were no treatment or no discontinuation of immunosuppressant? Especially for those treatment lasted 7 to 8 weeks. The author should discuss it in detail in the text.

2) Furthermore, there is a HEV pig model study proved that the immunocompromised conditions induced by the immunosupressants can cause a chronic infection of HEV (PNAS July 3, 2017 114 (27) 6914-6923). It supports the conclusion the authors make from the clinical data in this study. The authors may want to discuss it in the text.

Author Response

Reviewer 3:
Comments and Suggestions for Authors
In order to investigate the risk of chronic viral replication in immunocompromised patients other than transplant recipients and HIV infected patients. The authors of this manuscript examined the course of HEV infection in a retrospective cohort from seven internal medicine and rheumatology centers across Europe between 2014 and 2016. The result showed that 7 out of 21 rheumatology and internal medicine patients with HEV infection developed chronic HEV infection. They concluded that internal medicine/rheumatology patients, including patients treated with DMARDs, are at risk of developing chronic hepatitis E. Although there were publications on the same topic, this manuscript utilized more data and presented a more convincible conclusion. The study were careful carried out, and the manuscript is well written. However, I have two comments would like the authors to address.

è We are really grateful for the helpful comments of the reviewer. It was a good suggestion to highlight the difference between our present study and the previous study performed by Bauer et al..  

1) The Ref. 7 in this manuscript, suggested that there is no risk of chronicity of this infection in patients with inflammatory arthritis treated with immunosuppressants (Bauer H, et al. Medicine (Baltimore). 2015;94(14):e675.), which is conflict with the conclusion of this study. It is noteworthy that the outcome of the HEV infection in the previous study is under the interference of treatment. How long had the HEV infection been before the diagnosis? What happen if there were no treatment or no discontinuation of immunosuppressant? Especially for those treatment lasted 7 to 8 weeks. The author should discuss it in detail in the text.

è Good point. We discussed this in more detail.

2) Furthermore, there is a HEV pig model study proved that the immunocompromised conditions induced by the immunosupressants can cause a chronic infection of HEV (PNAS July 3, 2017 114 (27) 6914-6923). It supports the conclusion the authors make from the clinical data in this study.
The authors may want to discuss it in the text

è This anmímal study indead supports our observations. We cited this paper and mentioned it. Thanks.